# The Effects of Mycovirus BmPV36 on the Cell Structure and Transcription of *Bipolaris maydis*

**DOI:** 10.3390/jof10020133

**Published:** 2024-02-06

**Authors:** Yajiao Wang, Qiusheng Li, Yuxing Wu, Sen Han, Ying Xiao, Lingxiao Kong

**Affiliations:** Institute of Plant Protection, Hebei Academy of Agricultural and Forestry Sciences, Baoding 071000, Chinaalidd@163.com (Q.L.); wyx1209@163.com (Y.W.); hansenmayun@sina.com (S.H.); 13903369080@163.com (Y.X.)

**Keywords:** *Bipolaris maydis*, mycovirus, *Bipolaris maydis* partitivirus 36, transmission electron microscopy, transcriptome

## Abstract

*Bipolaris maydis* partitivirus 36 (BmPV36) is a mycovirus that can significantly reduce the virulence of the host *Bipolaris maydis*, but its hypovirulence mechanism is not clear. To investigate the response of *B. maydis* to BmPV36, the effects of BmPV36 on host cell structure and gene expression were studied via transmission electron microscopy and transcriptome sequencing using BmPV36-carrying and virus-free mycelium on the second and fifth culture. The results of transmission electron microscopy showed that the cell wall microfibrils of *B. maydis* were shortened, the cell membrane was broken, and membrane-bound vesicles and vacuoles appeared in the cells after carrying BmPV36. Transcriptome sequencing results showed that after carrying BmPV36, *B. maydis* membrane-related genes were significantly up-regulated, but membrane transport-related genes were significantly down-regulated. Genes related to carbohydrate macromolecule polysaccharide metabolic and catabolic processes were significantly down-regulated, as were genes related to the synthesis of toxins and cell wall degrading enzymes. Therefore, we speculated that BmPV36 reduces the virulence of *B. maydis* by destroying the host’s cell structure, inhibiting the synthesis of toxins and cell wall degrading enzymes, and reducing cell metabolism. Gaining insights into the hypovirulence mechanism of mycoviruses will provide environmentally friendly strategies for the control of fungal diseases.

## 1. Introduction

Maize is one of the most important staple crops worldwide, serving as a vital source of food, feed, and raw material for various industries [1]. With the rapid growth of demand in the field of renewable energy and deep processing, maize has become an important strategic resource in the 21st century. In 2022, China’s maize planting area was 43 million hectares, and the output was 277 million tons (https://data.stats.gov.cn/easyquery.htm?cn=C01, accessed on 16 December 2023), which was the largest food crop in China. Therefore, the production of maize plays a pivotal role in China’s national economy [2]. Maize is seriously threatened by a number of diseases throughout its growing period. In recent years, with the continuous large-scale planting of a single variety, the change in the cropping system, and the impact of global warming, the incidence of maize diseases has been continuously increasing, among which southern leaf blight of maize (caused by *Bipolaris maydis*) stood out as a significant concern [3]. The symptoms of southern leaf blight of maize first appeared in the lower leaves of maize and gradually expanded to the upper leaves. With the development of the disease, obvious spots are gradually formed, which affects the photosynthesis of the plants, leads to leaf dead stalk collapse, and seriously affects the growth and production of maize [4].

Traditionally, the application of disease-resistant varieties [5] and chemicals [6] is the major practice for controlling the southern leaf blight of maize. However, breeding resistant varieties takes time, and pathogens mutate frequently, so it is difficult to select and promote resistant varieties. Chemical fungicides are effective but come with drawbacks like environmental pollution and the emergence of resistant pathogen strains. With the abuse of chemical pesticides, an increased level of resistance to fludioxonil, carbendazim, and thiophanate-methyl appeared in *B. maydis* [6,7]. As a result, there has been increasing interest in exploring effective and environmentally friendly alternative methods for controlling this disease. Biological control has a wide application prospect with the advantages of low pollution, strong specificity, and environmental friendliness, which can reduce the application number of chemical pesticides and have a wide application prospect. Therefore, it is very necessary to carry out in-depth research on the development of biocontrol resources and the biocontrol mechanism of southern leaf blight of maize.

Mycoviruses are viruses that specifically infect fungi, including many plant-pathogenic fungi [8]. After infecting the host, most mycoviruses form a symbiotic relationship with the host and have no significant effect on the host phenotype. However, some mycoviruses can reduce the host growth rate, sporulation, and toxin production, resulting in a decrease in the host’s virulence, thus offering an environmentally friendly and sustainable means of disease control [9]. Recent studies have demonstrated the potential of mycoviruses in controlling a wide range of fungal diseases affecting various crops. *Fusarium graminearum* hypovirus 1 (FgHV1) could significantly reduce the disease severity of wheat *Fusarium* head blight [10]. Via stereoscopic microscope observation, it was found that *Botrytis cinerea* hypovirus 1 (BcHV1) and *Botrytis cinerea* fusarivirus 1 (BcFV1) could inhibit the formation of host infection cushion and inhibit the infection of *Botrytis cinerea* to onion [11]. *Sclerotinia sclerotiorum* hypovirulence-associated DNA virus 1 (SsHADV-1) could reduce the host virulence and could be transmitted in vitro. The application of mycelia fragments of *Sclerotinia sclerotiorum* containing SsHADV-1 in the field flowering period could successfully inhibit the occurrence of rapeseed stem rot and increase the yield of rape [12]. Understanding the mechanisms of hypovirulence of mycoviruses contributes to the development of sustainable and environmentally friendly biological control strategies.

At present, limited research has been conducted on mycoviruses associated with *B. maydis*. The mycovirus *Bipolaris maydis* partitivirus 1 was identified in *B. maydis* for the first time in 2017 [13], *Bipolaris maydis* botybirnavirus in 2018 [14], and *Helminthosporium victoriae* virus 190S in 2020 [15]. But the biological characteristics and interaction mechanisms with the host of these viruses are unclear. We studied the diversity of mycoviruses in *B. maydis* in Hebei Province, China, and found a hypovirus, which was 98.7% similar to *Bipolaris maydis* partitivirus 1 after sequencing. It was hence named *Bipolaris maydis* partitivirus 36 (BmPV36), but the hypovirulence mechanism of BmPV36 is unclear [16]. The aim of this study was to determine the effects of BmPV36 on biological characteristics (growth rate, sporulation, and virulence) of *B. maydis*; the effect of BmPV36 on the subcellular structure of *B. maydis*; the effect of BmPV36 on the transcription of *B. maydis* gene. Therefore, it was important to clarify the hypovirulence mechanism of BmPV36 and provide a scientific basis for the biological control of southern leaf blight of maize.

## 2. Materials and Methods

### 2.1. The Growth Rate, Sporulation, and Virulence of B. maydis Strains

The *B. maydis* strain BM36 carrying hypovirus BmPV36 was isolated from maize leaf with southern leaf blight symptoms on maize in Hebei Province, China. The virus-free strain BM36-Non is an isogenic strain that was derived from strain BM36 by protoplast isolation and regeneration. *B. maydis* strain BM36 and BM36-Non were cultured on potato dextrose agar (PDA) dish, respectively, at 25 °C in the dark for 5 d for four repetitions. The colony diameter was measured using the cross-crossing method, and the colony growth rate was calculated. Each dish of conidia was washed with 20 mL of sterile water to make a spore suspension, and the spore production was counted using a hemocytometer.

The virulence of *B. maydis* strains was evaluated on maize leaves. *B. maydis* strain BM36 and BM36-Non were cultured in 50 mL liquid potato dextrose broth (PDB), respectively, for five days at 25 °C in a shaker. The spores were harvested and resuspended with sterilized water to a concentration of 10^4^ spores/mL. Ten maizes (variety is Zhengdan 958) were seeded into a 12 cm pot; when maize plants were at the 4–5 leaves stage, spore suspension was prepared on the leaf surface, and each strain was sprayed on 6 pots as 6 repetitions; ddH_2_O was sprayed as a negative control. After spraying, maizes were incubated at 90% humidity and 35 °C for 24 h. Then, plants were grown in a greenhouse at 30 °C with 14 h of fluorescent light and 10 h of darkness for 7 d. The severity of southern leaf blight of maize was scored on 1, 3, 5, 7, and 9 scales (no symptoms, 1–10%, 11–25%, 26–50%, 51–75%, and 76–100% leaf area covered with speck). The disease index was then calculated as ∑ (Number of diseased plants × Corresponding score)/(Total number of plants × Highest score) × 100%.

### 2.2. Evaluation of the Cellular Morphology of B. maydis Strain Mycelia via Transmission Electron Microscopy (TEM)

The *B. maydis* strain BM36 was cultured on PDA, and the mycelium was collected after 2 d (early growth stage; no sporulation) and 5 d (sporulation stage). The mycelium samples were fixed with 2.5% glutaraldehyde at room temperature for 2 h, rinsed with 0.1 M phosphate-buffered saline (pH 7.4) three times, and then fixed with 1% osmic acid at 4 °C for 2 h. Then, the samples were dehydrated according to 50%, 70%, 80%, 85%, 90%, 95%, and 100% ethanol, permeated successively with 2:1 and 1:1 acetone and epoxy resin as a penetrant, and embedded in ethoxyline resin. The mycelium of strain BM36-Non was collected in the same way and treated as the control. Ultrathin sections were made using the EM UC7 diamond microtome (Leica Microsystems, Wetzlar, Germany) and placed on a formvar-coated copper grid, stained with lead and uranium, and observed for the damage to cell morphology of *B. maydis* by virus-carrying using a transmission electron microscopy Tecnai G2-20 TWIN (FEI Company, Hillsobro, OR, USA) under a 200-kV accelerating voltage.

### 2.3. Effect of BmPV36 on the Cell Membrane Integrity of B. maydis Mycelium

The *B. maydis* strain BM36 was cultured on PDA, and the mycelium was collected after 2 d (early growth stage; no sporulation) and 5 d (sporulation stage). Then, the collected mycelium was immersed in 20 µM propidium iodide (PI) in the dark for 10 min. The mycelium of strain BM36-Non was collected in the same way and treated as the control. PI can penetrate the damaged membrane of cells to stain the DNA. The mycelium was observed under a fluorescence microscope (Olympus Corporation, Tokyo, Japan).

### 2.4. RNA Extraction and Sequencing

BM36 was cultured on PDA, and the mycelium was collected after 2 d (BM2) and 5 d (BM5) cultivation, four repetitions. The mycelium of strain BM36-Non was collected after 2 d (BM-Non2) and 5 d (BM-Non5) cultivation as the control. Total RNA was extracted from mycelium using an RNA extraction kit (Invitrogen, Waltham, MA, USA) according to the manufacturer’s instructions. Then, RNA quality was determined by 2100 Bioanalyser (Agilent Technologies, Santa Clara, CA, USA) and quantified using a Qubit fluorometer (Life Technologies, Carlsbad, CA, USA). Only high-quality RNA sample (OD 260/280 = 1.8~2.2, OD 260/230 ≥ 2.0, RIN ≥ 8.0, 28 S: 18 S ≥ 1.0, >1 μg) was used to construct sequencing library. RNA purification, reverse transcription, library construction, and sequencing were performed at Shanghai Majorbio Bio-pharm Biotechnology Co., Ltd. (Shanghai, China) according to the manufacturer’s instructions (Illumina, San Diego, CA, USA). The transcriptome library was prepared with TruSeq TM RNA sample preparation Kit (Illumina, San Diego, CA, USA) using 1 μg of total RNA. Messenger RNA was isolated and then fragmented. cDNA was synthesized with random hexamer primers by a SuperScript double-stranded cDNA synthesis kit (Invitrogen, Carlsbad, CA, USA). Then, the synthesized cDNA was subjected to end-repair, phosphorylation, and “A” base addition according to Illumina’s library construction protocol. A library with a 300 bp cDNA target fragment was selected on 2% Low Range Ultra agarose, and 15 PCR cycles were amplified using Phusion DNA polymerase (NEB). After quantified by TBS380, the paired-end RNA-seq sequencing library was sequenced with the Illumina NovaSeq 6000 sequencer (2 × 150 bp read length).

### 2.5. Differential Expression Analysis

The raw paired-end reads were trimmed and quality controlled by fastp v. 1.8.4 (https://github.com/OpenGene/fastp, accessed on 16 October 2023) [17]. Then, the clean reads were separately aligned to the reference genome using HISAT2 v. 2.1 (http://ccb.jhu.edu/software/hisat2/index.shtml, accessed on 16 October 2023) [18]. The mapped reads were assembled by StringTie v. 2.2.1 (https://ccb.jhu.edu/software/stringtie/, accessed on 18 October 2023) [19]. In order to identify differential expression genes between two different samples, the expression level of each transcript was calculated according to the transcripts per million reads (TPM) method. Gene abundance was quantified by RSEM [20]. Essentially, differential expression analysis was performed by DESeq2 [21]. DEGs with |log2FC| ≥ 1 and FDR < 0.05 (DESeq2) or FDR < 0.001 (DEGseq) were considered to be significantly differentially expressed genes. In addition, functional-enrichment analysis, including GO and KEGG, was performed to identify which DEGs were significantly enriched in GO terms and metabolic pathways at Bonferroni-corrected *p*-value < 0.05 compared with the whole-transcriptome background. GO functional-enrichment, and KEGG pathway analysis were carried out by Goatools and Python script, respectively.

### 2.6. Real-Time Quantitative Reverse Transcription PCR Analysis

First-strand cDNA was generated from 3 µg total RNA, which was the same RNA used for RNA-Seq analysis. cDNA was synthesized with an oligo d(T) primer using the reverse transcription reagent kit (Applied Biosystems, Carlsbad, CA, USA) for q-PCR. Primers for the target genes are listed in Appendix A. Real-time quantitative reverse transcription PCR (RT-qPCR) was performed using q-PCR SYBR Green mix in 20 µL reactions. Each reaction consisted of 20 ng cDNA, 0.5 µL of each primer, and 10 µL master mix. PCR reactions were carried out under the following conditions: 3 min at 95 °C, 40 cycles of 15 s at 95 °C, 15 s at 57 °C, and 20 s at 72 °C. Following this, melting curve analysis of amplification products was performed at the end of each PCR reaction, which verified the presence of a specific product for 2 min at 16 °C [22]. The actin gene serves as an internal reference gene, as it is expressed stably in the nucleus. mRNA relative expression level changes were quantified using a comparative CT method (ΔΔCT) using the formula 2^−ΔΔCt^ [23]. Data were presented as mean ± SD and analyzed using SPSS (Version 21.0) software. Data were further evaluated using Duncan’s new multiple-range test. *p* < 0.05 was considered to indicate statistical significance [24].

## 3. Results

### 3.1. The Effect of BmPV36 on the Growth Rate, Sporulation, and Virulence of B. maydis

After 5 days of culture, the colony morphology, growth rate, and spore production of BM36 (*B. maydis* with BmPV36) and BM36-Non (virus-free *B. maydis*) were detected, and the results show that the BM36 colony was gray, and the mycelium was fluffy (Figure 1a). But the colony of BM36-Non was black-gray, and the mycelium was relatively dense (Figure 1b). After 5 days of culture, the colony diameter and spore concentration of BM36 were 5.35 cm and 7.53 × 10^4^ spores/mL, respectively, while the colony diameter and spore concentration of BM36-Non were 5.55 cm and 6.23 × 10^5^ spores/mL, respectively (Appendix A). There was no significant difference in growth rate between BM36 and BM36-Non. However, the sporulation of *B. maydis* decreased significantly after carrying BmPV36. Spores of BM36, BM36-non and ddH_2_O were sprayed on the surface of maize leaves to test the virulence, and the disease index was 25.35, 65.25 and 0, respectively (Figure 1c–e), indicating that the virulence of *B. maydis* was reduced after carrying BmPV36.

### 3.2. The Effect of BmPV36 on the Cell Structure of B. maydis

The effect of BmPV36 on the cell structure of *B. maydis* was observed by transmission electron microscopy. On the second day (Figure 2a,e) and fifth day of culture (Figure 2b,f), BM36-Non cell had a complete and regular structure, clear outline, normal cell wall structure, uniform thickness, visible cell wall layer, long and dense microcilia on the outside of the cell wall, and regular mitochondria and nuclei were visible inside the cell. Vacuoles appeared in the cell on the fifth day but occupied a small area. When carrying BmPV36, the microfibrils on the outer layer of the cell wall are short and few, the cell membrane is damaged and blurred, and the organelles are not clear. On the second day of culture (Figure 2c,g), vesicles appeared in the cells, and virions were distributed in the vesicles. On the fifth day of culture (Figure 2d,h), a large vacuole appeared in the cells, accounting for about 80% of the cell volume. Virions were distributed in the vacuole, and the edge of the vesicle gradually fused with the cytoplasm.

Furthermore, the membrane integrity of *B. maydis* mycelium was examined by fluorescence observation. On the second day (Figure 3c) and fifth day of culture (Figure 3d), the mycelium of BM36 was stained fluorescent red, whereas the red fluorescence could not be observable in the mycelium of BM36-Non cultured on the second day (Figure 3a) and fifth day (Figure 3b). The results showed that the membrane integrity of *B. maydis* was destroyed after carrying BmPV36.

### 3.3. Overview and Validation of Differentially Expressed Genes (DEGs)

Four samples, including BM2, BM5, BM-Non2, and BM-Non5, were sequenced. After filtration of low-quality reads and adapter sequences, a total of 47,422,239 (BM2), 41,637,599 (BM5), 47,103,270 (BM-Non2), and 44,169,727 (BM-Non5) clean reads of each sample was obtained, which accounted for 97.61%~97.71% of the original data. Among them, 39,781,816~45,181,476 clean reads were compared to the genome of *B. maydis*, accounting for 94.66%~95.54% (Table 1). These results indicated that the sequencing data were reliable for further analysis. Principal component analysis (PCA) was used to analyze the differences between the samples (Appendix A). The first principal component (PC1) and the second principal component (PC2) represent the difference of 57.88% and 14.62% of the samples. The samples in each group were clearly differentiated, and the repetitions between the groups were able to cluster together. In summary, the similarity of sequencing samples within the group was good, and the difference between the groups was obvious. Follow-up analysis of gene expression differences can be performed.

Differentially expressed genes (DEGs) were identified with a threshold of a 2-fold change in expression relative to the virus-free sample and a *p*-value less than 0.05. Compared with BM-Non2, there were 2002 differential expressed genes in BM2, of which 1141 were up-regulated and 861 were down-regulated (Figure 4a). Compared with BM-Non5, there were 1531 differential genes in BM5, of which 983 were up-regulated and 548 were down-regulated (Figure 4b). A total of 2911 genes were differentially expressed at both time points; 622 genes (21.37%) were differentially expressed on both the second day and fifth day. Moreover, 1380 (47.41%) and 909 genes (31.23%) were identified as differentially expressed on the second day and fifth day, respectively (Figure 4c). The lists of differentially expressed genes according to those down-regulated vs. those upregulated were analyzed (Figure 4d,e). Among the down-regulated genes, more genes were induced on the second day (749 genes) than on the fifth day (436 genes), and only 112 (8.64%) genes were induced on both the second day and fifth day (Figure 2e).

Among the up-regulated genes, more genes were induced on the second day (929 genes) than on the fifth day (771 genes), and only 212 (11.09%) genes were induced on both the second day and fifth day.

### 3.4. GO Annotation and Enrichment Analysis of the DEGs

We subjected a total of 2911 differentially expressed genes to gene ontology (GO) annotation to gain insight into their functional classifications. To obtain insight into essential gene functions regulated after carrying BmPV36, we conducted a GO enrichment analysis of the differentially expressed genes. The results showed that the genes of *B. maydis* had significant differences in cell components, molecular functions, and biological processes (Figure 5 and Appendix A). The most significantly enriched GO was the intrinsic component of the membrane and integral component of the membrane, which were up-regulated on the second day and fifth day of culture. In molecular functions, metabolic processes such as carbohydrate, macromolecule, polysaccharide metabolic, and catabolic processes were down-regulated on the second day and 5 h. In biological processes, catalytic activity was specially down-regulated on the second day, and copper ion homeostasis and transporter activity were specially down-regulated on the fifth day.

### 3.5. Influence of BmPV36 on Fungal Host Gene Expression Related to Metabolic Process

Carrying BmPV36 significantly affected the expressions of genes related to metabolic pathways. GO diagram showed that genes associated with a variety of metabolic pathways were significantly down-regulated on the second day and fifth day of culture (Figure 6). For example, over-represented GO terms included carbohydrate metabolic process (GO:0005975); cellular carbohydrate metabolic process (GO:0044262); cellular carbohydrate catabolic process (GO:0044275); macromolecule catabolic process (GO:0009057); polysaccharide metabolic process (GO:0005976); cellular polysaccharide catabolic process (GO:0044247); glucan metabolic process (GO:0044042); and cellular glucan metabolic process (GO:0006073). However, genes involved in the secondary metabolic process were only significantly down-regulated on the fifth day and had no significant differences on the second day. The down-regulated GO terms of secondary metabolic process included secondary metabolic process (GO:0019748); secondary metabolite biosynthetic process (GO:0044550); toxin metabolic process (GO:0009404); toxin biosynthetic process (GO:0009403); mycotoxin metabolic process (GO:0043385); and mycotoxin biosynthetic process (GO:0043386). Ophiobolin is the mycotoxin of *B. maydis* [16]. After carrying BmPV36, genes of *B. maydis* related to two key enzymes in ophiobolin synthesis were significantly down-regulated; they were geranylgeranyl diphosphate synthase (COCC4DRAFT_123791) and mevalonate kinase (COCC4DRAFT_50090) (Appendix A).

### 3.6. Influence of BmPV36 on Fungal Host Gene Expression Related to Cell Wall and Membrane

Carrying BmPV36 significantly affected the expressions of genes related to cell walls and membranes. The GO diagram showed that genes associated with the cell wall were significantly down-regulated on the second day and fifth day (Figure 6), for example, the cell wall macromolecule metabolic process (GO:0044036) and the cell wall polysaccharide metabolic process (GO:0010383). However, membrane-related genes were significantly up-regulated on the second day and fifth day (Appendix A), which included the membrane (GO:0016020); the intrinsic component of the membrane (GO:0031224); the intrinsic component of the plasma membrane (GO:0031226); the anchored component of the membrane (GO:0031225); the integral component of the membrane (GO:0016021); and the integral component of the plasma membrane (GO:0005887).

### 3.7. Influence of BmPV36 on Fungal Host Gene Expression Related to Transporter

Transporter activity was one of the significantly over-represented GO terms with respect to molecular function. GO diagram showed that genes associated with ion, carbohydrate, and amide transmembrane transporter activity were significantly down-regulated on the second day and fifth day (Figure 7). The down-regulated ion transmembrane transporter activity included anion transmembrane transporter activity (GO:0008509); cation transmembrane transporter activity (GO:0008324); inorganic anion transmembrane transporter activity (GO:0015103); and inorganic cation transmembrane transporter activity (GO:0022890). The down-regulated carbohydrate transmembrane transporter activity included secondary active transmembrane transporter activity (GO:0015291) and inorganic phosphate transmembrane transporter activity (GO:0005315). The down-regulated amide transmembrane transporter activity included urea transmembrane transporter activity (GO:0015204); peptide transmembrane transporter activity (GO:1904680), and oligopeptide transmembrane transporter activity (GO:0035673).

### 3.8. Validation of mRNA Transcriptome Analysis Data by RT-qPCR

Representative fungal genes that showed significant gene expression were listed in Appendix A. After carrying BmPV36, genes of *B. maydis* encoding cell wall degrading enzyme were up-regulated, such as cellulase D (COCC4DRAFT_29453, COCC4DRAFT_140403, COCC4DRAFT_88117, COCC4DRAFT_124946, and COCC4DRAFT_53511), pectinesterase (COCC4DRAFT_175201), and cutinase (COCC4DRAFT_18807). In addition, 18 MFS transporter-related genes and 2 toxin synthesis-related genes (COCC4DRAFT_123791 and COCC4DRAFT_50090) were also significantly down-regulated. To confirm RNA-Seq results, we selected eight genes and verified mRNA transcriptome analysis data by RT-qPCR. Results showed that compared with virus-free *B. maydis*, genes of BM2 encoding cellulase D (COCC4DRAFT_29453, COCC4DRAFT_53511), pectinesterase (COCC4DRAFT_175201), and cutinase (COCC4DRAFT_18807) were down-regulated by 50%, 47.5%, 90%, and 52.2%, while those of BM5 were down-regulated by 71.23%, 56.89%, 67.36, and 65%. Genes of BM2 encoding MFS domain-containing protein (COCC4DRAFT_82543 and COCC4DRAFT_80536) were down-regulated by 76.6% and 52.2%, while those of BM5 down-regulated by 79.73% and 81.82%. Genes of BM2 encoding toxin synthesis-related genes (COCC4DRAFT_123791 and COCC4DRAFT_50090) were down-regulated by 75% and 45%, while those of BM5 were down-regulated by 70.27% and 80% (Figure 8b). The results of RT-qPCR were highly consistent with those of RNA-Seq (Figure 8a).

## 4. Discussion

### 4.1. BmPV36 Changed the Cell Wall Structure of B. maydis and Shortened the Microfibrils

Fungal cell wall microfibrils are fine filamentous structures composed of polysaccharides and proteins, which are an important part of the fungal cell wall and play a key role in the structure and function of the cell wall [25]. (1) Maintaining cell shape: microfibrils are support structures in the fungal cell wall that give the cell wall mechanical strength and help maintain the shape of the cell [26]. (2) Responsible for material transport: microfibrils can act as a material transport channel, facilitating the exchange of substances between different areas within the cell wall [27]. (3) Participation in immune response: glycans and proteins on microfibrils can participate in the immune response of fungal cells [28]. After carrying BmPV36, the relative genes of the cell wall macromolecule metabolic process (GO:0010383), cell wall organization (GO:0071555), and structural constituent of the cell wall (GO:0005199) were down-regulated, and the microfibrils of the cell wall became shorter. Microfibrils are often the first barrier between fungal cells and the external environment, and sparse microfibrils make it easier for viruses to invade fungal cells. Microfibrils can be involved in the immune response of fungal cells, triggering antiviral defense mechanisms by interacting with viruses [28]. Sparse microfibrils may have attenuated this immune response, making fungal cells more susceptible to viral infection.

### 4.2. Effect of BmPV36 on the Cell Membrane of B. maydis

After carrying BmPV36, the relative genes of the membrane intrinsic component and integral component were significantly up-regulated. Electron microscope observation showed that the cell membrane of *B. maydis* was damaged after infection by BmPV36. Therefore, we speculated that in order to restore the integrity of the damaged cell membrane, membrane-related genes were up-regulated. Similarly, the infection of mycoviruses chrysovirus 1 (BdCV1) and partitivirus 1 (BdPV1) significantly up-regulated the cell membrane-related genes of *Botryosphaeria dothidea* [23]. Another reason for the up-regulated membrane-related genes may be that virus infection induced the host to produce a large number of membrane-related vacuoles and vesicles. It was observed that membrane-bound vacuoles and vesicles with virions were found by TEM in the cells of *B. maydis* with the infection of BmPV36. Similar membrane structures were also examined in the *Sclerotium rolfsii* infected with mycoviruses SsHV2 [29] and in the *Sclerotium sclerotiorum* infected with mycovirus SsHV2L [30]. Membrane-bound vesicles play a variety of roles in virus-infected cells, including intracellular viral replication, transport, and immune response. After infecting cells, viruses can induce the host cell to form specific replication vesicles, which help provide a specific environment for the virus to multiply and evade the immune response in the host cell [31].

### 4.3. BmPV36 Regulated the Transmembrane Transport of B. maydis

Mycoviruses can promote their replication and spread by regulating the host transmembrane transport [32]. Via high-throughput sequencing, *Sclerotinia sclerotiorum* hypovirus 2-L (SsHV2-L) was found to affect the expression of 985 nucleodiscus genes, more than 100 of which were related to metabolism, carbohydrate, and lipid transport of nucleodiscus [33]. Early transmembrane transporter-related genes of *Fusarium graminis* infected with *F. graminearum* virus 1 strain-DK21 (FgV1-DK21) were also significantly down-regulated [34]. After carrying BmPV36, *B. maydis* had many down-regulated genes related to transmembrane transport, such as ion, carbohydrate, and amide transmembrane transporter, which are important processes closely related to basic life activities. The transmembrane transport system of fungi is mainly composed of two types: ABC (ATP-Binding cassette transporter) system and MFS (major facilitator superfamily) transport system. The ABC transport system by hydrolyzing ATP to produce energy for material transport; the MFS transport system relies on chemical osmosis formed by ion gradient to transport substances [35]. Transporters are important carriers of transport systems and have important functions. Deletion of an MFS transporter-like gene in *Cercospora nicotianae* reduces cercosporin toxin accumulation and fungal virulence [36]; the MFS transporter of *Botrytis cinerea* is a virulence factor that increases tolerance to glucosinolates [37]; and the MFS transporter is involved in the self-defense of *Fusarium graminis* against DON [38]. Eighteen MFS transporter-related genes were significantly down-regulated by *B. maydis* after carrying BmPV36, but the function of these transporters needs further study.

### 4.4. BmPV36 Reduced the Virulence of B. maydis

Fungal virulence is a very complex process. Mycotoxins, hydrolases, and small secreted proteins are important pathogenic factors which can significantly affect the virulence of fungi [39]. In our study, genes related to mycotoxin synthesis and metabolism were significantly down-regulated after BmPV36 infection (GO:0009403; GO:0009404; GO0043385; and GO004386). Ophiosporin is a non-host-specific toxin produced by filamentous fungi and an important pathogenic factor. Its synthesis pathway is derived from the Mevalonate pathway (MVA pathway) [40]. It can also change the permeability of cell membranes, reduce the fixation of carbon dioxide in photosynthesis, and make plant leaves produce brown spots. *B. maydis* can secrete ophiobolin [41]. In our study, it was found that the genes related to two key enzymes in ophiobolin synthesis were significantly down-regulated; they were geranylgeranyl diphosphate synthase (COCC4DRAFT_123791) and mevalonate kinase (COCC4DRAFT_50090). The cell wall degrading enzymes secreted by fungi are important factors in successfully infecting plants. The most common cell wall degrading enzymes of plant-pathogenic fungi include cellulase, hemicellulase, pectinase, polyphenol oxidase, and some proteases [42]. After BmPV36 infection, the genes related to cellulase pectinesterase and cutinase synthesis were significantly decreased in *B. maydis*. Therefore, we speculated that BmPV36 could reduce the virulence of *B. maydis* by inhibiting the synthesis of *B. maydis* toxin and cell wall degrading enzyme. In addition, the first and key step of fungal infection is adhesion and invasion. Microfibrils of fungal cell walls can bind to molecules on the surface of host cells, thus promoting the adhesion of fungi to the host [43]. After BmPV36 infection, the microfibrils of *B. maydis* became shorter and smaller, which could inhibit the adhesion of *B. maydis* to the host and finally inhibit the infection.

## Figures and Tables

**Figure 1 jof-10-00133-f001:**
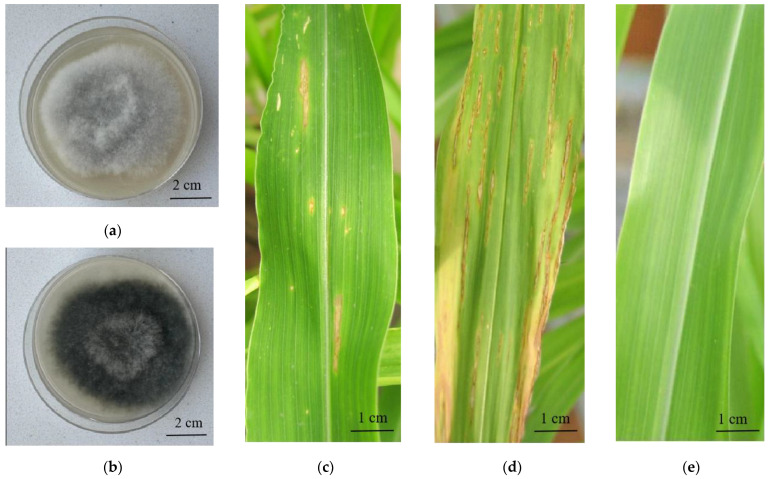
Characteristics and pathogenicity of virus-free *B. maydis* and BmPV36-carrying *B. maydis*. Characteristics of BmPV36-carrying *B. maydis* (**a**) virus-free *B. maydis* (**b**); Pathogenicity of BmPV36-carrying *B. maydis* (**c**), virus-free *B. maydis* (**d**), and negative control (dd H_2_O) (**e**).

**Figure 2 jof-10-00133-f002:**
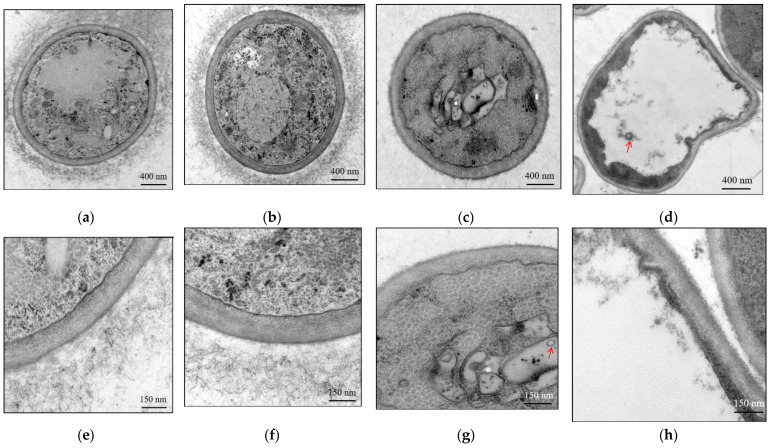
Microscopic observation cell ultrastructure of fungal hyphae from virus-free *B. maydis* and BmPV36-carrying *B. maydis*. Virus-free *B. maydis* cultured on PDA for 2 d (**a**,**e**) and 5 d (**b**,**f**); BmPV36-carrying *B. maydis* cultured on PDA for 2 d (**c**,**g**) and 5 d (**d**,**h**); The red arrows point to the virus particles.

**Figure 3 jof-10-00133-f003:**
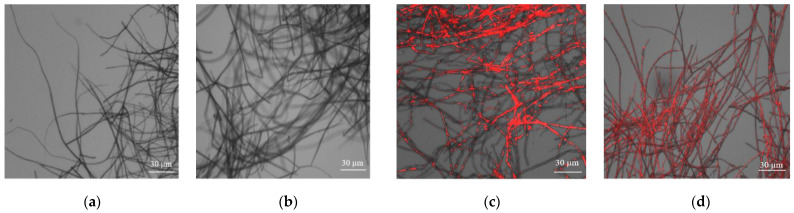
Propidium iodide (PI) staining to determine the mycelial cell membrane integrity of virus-free *B. maydis* and BmPV36-carrying *B. maydis*. Virus-free *B. maydis* cultured on PDA for 2 d (**a**) and 5 d (**b**); BmPV36-carrying *B. maydis* cultured on PDA for 2 d (**c**) and 5 d (**d**).

**Figure 4 jof-10-00133-f004:**
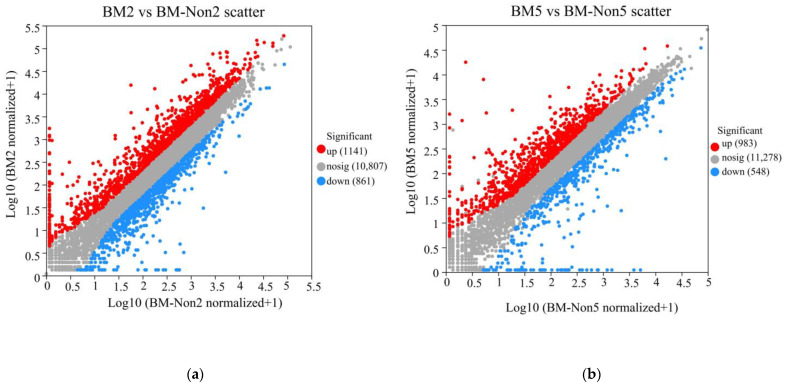
Differentially expressed genes of BmPV36-carrying *B. maydis* compared to virus-free *B. maydis*. Scatterplots on the second day (**a**) and fifth day (**b**) of culture. The Venn diagram illustrates the total number of genes that were significantly differentially expressed on the second day and fifth day (**c**) of culture, down-regulated genes (**d**), and up-regulated genes (**e**).

**Figure 5 jof-10-00133-f005:**
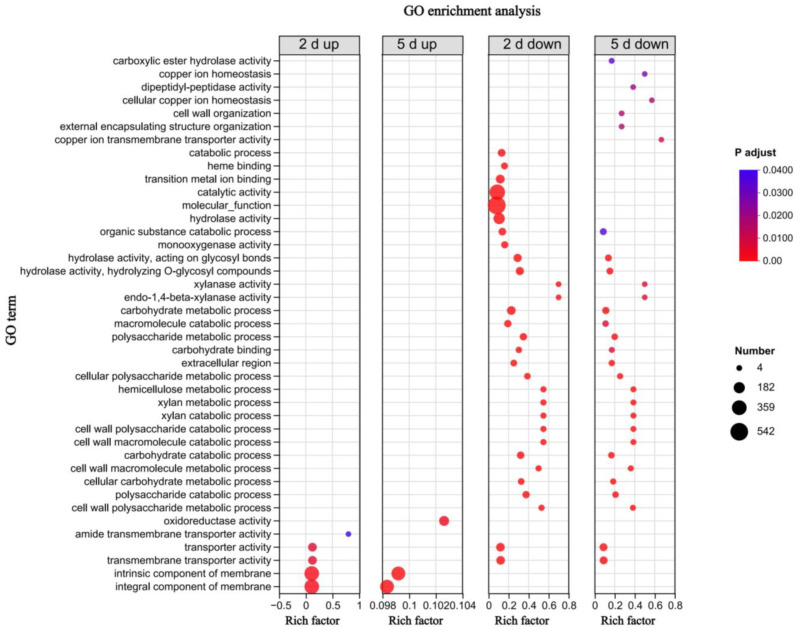
GO enrichment analysis of differentially expressed genes of BmPV36-carrying *B. maydis* cultured on the second day and fifth day of culture. The Y-axis is the GO term, the X-axis is the rich factor, the bubble size represents the number of genes, and the color is the corrected *p*-value.

**Figure 6 jof-10-00133-f006:**
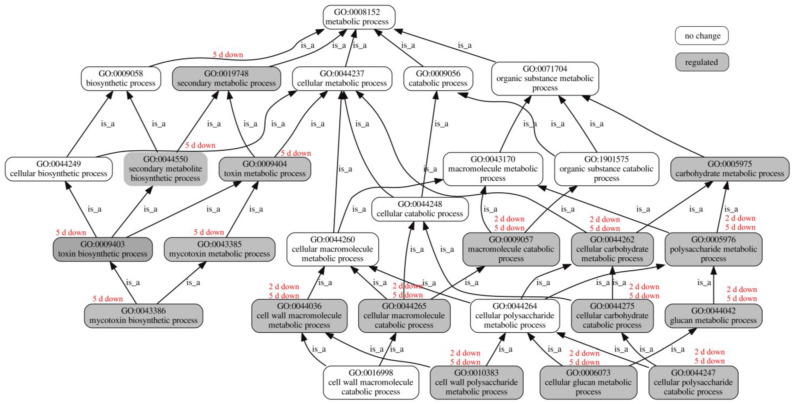
GO diagram of significantly over-represented GO terms in biological processes associated with metabolism.

**Figure 7 jof-10-00133-f007:**
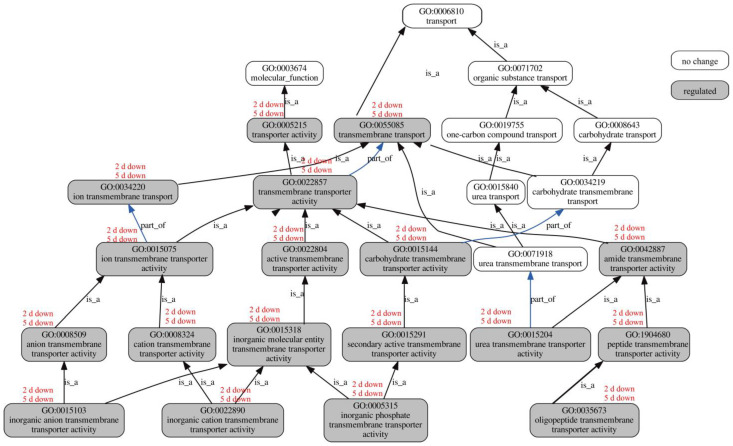
GO diagram of significantly over-represented GO terms in transporter activity.

**Figure 8 jof-10-00133-f008:**
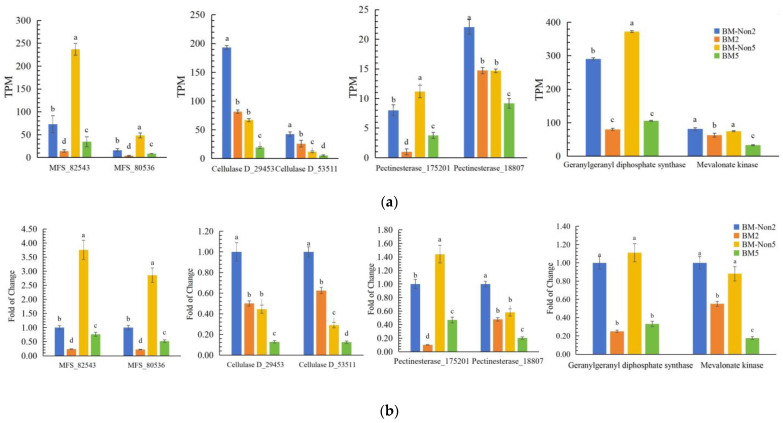
The expression of eight differentially expressed genes. (**a**): Transcripts per kilobase million (TPM) of eight differentially expressed genes detected by RNA-sequencing; (**b**): Fold change of eight differentially expressed genes detected by RT-qPCR. Different letters above bars differed signifcantly (*p* < 0.05) according to Duncan’s test.

**Table 1 jof-10-00133-t001:** Summary of sequencing data.

	Raw Reads	Clean Reads	Clean Reads Ratio (%)	Genome Mapping Reads	Genome Mapping Reads Ratio (%)
BM2	48,718,366	47,422,239	97.61	45,181,476	95.28
BM5	42,356,456	41,637,599	97.69	39,781,816	95.54
BM-Non2	47,605,831	47,103,270	97.67	44,587,291	94.66
BM-Non5	44,715,156	44,169,727	97.71	41,940,611	94.95

Note: Clean reads ratio: Clean reads/raw reads; Genome mapping reads ratio: Genome mapping reads/raw reads; BM2: BmPV36-carrying *B. maydis* cultured for 2 d; BM5: BmPV36-carrying *B. maydis* cultured for 5 d; BM-Non2: Virus-free *B. maydis* cultured for 2 d; BM-Non5: Virus-free *B. maydis* cultured for 5 d.

## Data Availability

The data presented in this study are openly available in the NCBI Sequence Read Archive, with the accession number PRJNA1056555.

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
