# Peer review of "The Effects of Mycovirus BmPV36 on the Cell Structure and Transcription of Bipolaris maydis"

_jof, 2024, doi:10.3390/jof10020133_

Round 1

Reviewer 1 Report

Comments and Suggestions for Authors

The manuscript entitled " Effects of mycovirus BmPV36 on the cell structure and transcription of Bipolaris maydis" presents an analysis of the effect of a mycovirus of Bipolaris maydis. The author shows in this article that the BmPV36 reduces the virulence in maize plants.

Although the article provides valuable insights into how the reduction in Bipolaris maydis virulence is due to this mycovirus, there are some critiques and suggestions that could enhance the manuscript:

Main Criticisms

First, I believe the manuscript would benefit the Bipolaris maydis research community. However, there are several unknown aspects of how this mycovirus reduces its virulence that in my opinion are not clear with this work. I suggest that only a transcriptome is not enough to define a function of the mycovirus. Several changes in the transcriptome between infected and noninfected are due to an indirect effect of the mycovirus and not a direct effect.  

Second, I miss a section where the author clearly shows the presence or absence of this mycovirus or others in these fungal strains. Are there more mycoviruses in these strains? It is reported that several mycoviruses exhibit synergistic behaviour, thus this aspect should be relevant to gain insight into the effect of this partitivirus in a multi-infected strain.   

Third, in my opinion, TEM is not enough to confirm the effect on cell structure. A complementary experiment should be helpful to determinate some structure changes. Primarily, how do you confirm that the structure and virions that you describe are true? How do you know that virions are in the vesicles? A large presence of vesicles at 5 days is a fungal stress effect.

Optical microscopy using specific stains should be useful for confirming some of the assumptions reported using TEM.  In addition, the use of some disruptive chemicals (e.g. calcofluor white or Congo red) can also help confirm cell wall damage or dysfunction in the cell wall.

Regarding DEGs, the experiment is commendable and thoroughly descriptive; however, it highlights only a limited number of interesting findings. Furthermore, certain conclusions about changes in transcriptional regulations lack experimental confirmation.

Is the hypovirulence of BmPV36 exclusive of this strain studied or do other strains of B. maydis that are infected with this mycovirus show a similar phenotype?

A molecular characterisation of the partitivirus should be included. Understanding how the mycovirus expresses itself during infection or fungal development will significantly enhance the manuscript's impact post-publication.

Several issues need attention, particularly Figure 1, where improvement is needed. The figures' panels should be positioned directly above each image (a, b, ...). This will aid the reader, and incorporating a table with different phenotypic effects, error rates, and statistical analyses (e.g., sporulation, growth rates, disease index) from Table S2 into principal Fig. 1 is recommended. Include the number of biological replicates and consider a germination test to ensure the observed reduction in virulence is not due to this aspect.

For panels (c) and (d), although changes in virulence are visually clear, a graphical representation of the disease index over time with error rates and statistical analysis is missing. Objective representation using image analysis, such as infected area vs. healthy area in mm2, is preferred. Additionally, include negative controls in these experiments. Correct "d" to "days" in line 107.

Figure 2 requires improvement similar to Figure 1, with the panels above each picture. Descriptive figures without the use of panel letters are encouraged.

In Figure 7, clarify whether gene expression is presented as the Fold of Change. Confirm the calibrator used and provide references used in RT-qPCR, as this information is unclear in the Materials and Methods section. It is recommended to use a graphic per gene for better visibility of unrelated changes. Consider using gene names or functional names instead of exclusive gene IDs. Compare results of FC in RNA-seq vs. RT-qPCR to validate findings and address potential biases between the two experimental approaches.

Minor critiques:

Minor critiques include the manuscript lacking groundbreaking findings but offering informative responses to general questions about BmPV36's effects on B. maydis. Consider a computational pipeline figure for the full DE analysis. Clarify the use of "pathogenicity" and "virulence," ensuring consistent and accurate terminology. Streamline the discussion to be concise and focused.

In conclusion, while the manuscript provides valuable information, a major revision is recommended to address these issues and enhance overall manuscript quality.

Finally, I would like to express my sincere appreciation to the authors for their diligent efforts in conducting the research presented in the manuscript. Their commitment to advancing the understanding of BmPV36 impact on B. maydis is evident in the comprehensive experimental work and thorough descriptions provided. Despite the identified areas for improvement, the authors' dedication to their scientific inquiry is commendable. I am grateful for the opportunity to review their work and believe that with the suggested revisions, the manuscript has the potential to contribute significantly to the field.

Reviewer 2 Report

Comments and Suggestions for Authors

The study focuses on Bipolaris maydis partitivirus 36 (BmPV36), a mycovirus known to reduce the pathogenicity of its host. By employing transmission electron microscopy and transcriptome sequencing, the research investigates the impact of BmPV36 on the cell structure and gene expression of B. maydis. Results reveal notable changes in cell morphology, including shortened microfibrils and membrane damage, indicating the hypovirulent effects of BmPV36. Transcriptome analysis shows significant alterations in gene expression related to membrane function, carbohydrate metabolism, toxin synthesis, and cell wall degradation. The findings suggest that BmPV36 diminishes pathogenicity by disrupting cell structure, suppressing toxin and enzyme synthesis, and reducing cellular metabolism. This understanding contributes to developing eco-friendly strategies for fungal disease control.

The manuscript exhibits a commendable level of writing and organization. With substantial revisions, I recommend it for publication.

How about to change the title as follows.

Impact of the Mycovirus BmPV36 on Cellular Structure and Transcriptome in Bipolaris maydis

Minor comments

Line 10: "BmPV36-carrying and virus-free mycelium on the 2 d and 5 d culture." - It's better to mention "2nd day" and "5th day" for clarity.

Line 43: "times of breeding resistant varieties, the frequent mutation..." - It might be clearer to say "breeding resistant varieties takes time, and frequent mutations..."

Line 48: "come with drawbacks like environmental pollution..." - It should be "comes with drawbacks..."

Line 61: "sustainable means of disease control [9]." - It's good to provide a citation, but it might be more informative if you briefly mention what the citation supports.

Line 63: "Fusarium graminearum hypovirus 1 (FgHV1) was found in Fusarium graminearum, which can significantly reduce the disease severity..." - It's a bit unclear. Consider rephrasing for clarity.

Line 76: "first time [13]. Bipolaris maydis botybirnavirus [14] and Helminthosporium victoriae virus 190S [15] were subsequently found in B. maydis, but..." - The sentence structure is a bit confusing. Consider breaking it down for better clarity.

Line 82: "The aim of this study were to determine the effects..." - It should be "The aim of this study was to determine..."

Line 98: "pathogenicity and the disease index was 25.35 and 65.25 respectively(Fig. 1c, d)..." - The word "respectively" should be placed before the values. "The disease index was 25.35 and 65.25, respectively (Fig. 1c, d)."

- Line 330: Consider rephrasing "southern leaf blight of maize" to "southern leaf blight symptoms on maize."

- Line 336: "spore suspension was sprayed" should be "spore suspension was prepared."

- Line 341: Consider using "maize plants" instead of "maizes."

- Line 345: Consider rephrasing "of fuorescent light" to "of fluorescent light."

- Line 347: The formula for disease index calculation seems incomplete; please double-check and provide the correct formula.

- Line 361: "viral carrying" can be rephrased to "virus-carrying."

- Line 364: It should be "repetitions" instead of "repetitions."

- Line 393: It should be "significantly differentially expressed genes" instead of "significantly differ-ent expressed genes."

- Line 405: Consider rephrasing "PCR reactions were performed using" to "PCR reactions were carried out with."

Comments on the Quality of English Language

Minor editing of English language required.

Round 2

Reviewer 1 Report

Comments and Suggestions for Authors

The author's response comments have been provided in PDF file above in blue text. Thank you
